# Stressor-Induced Increases in Circulating, but Not Colonic, Cytokines Are Related to Anxiety-like Behavior and Hippocampal Inflammation in a Murine Colitis Model

**DOI:** 10.3390/ijms23042000

**Published:** 2022-02-11

**Authors:** Ross M. Maltz, Pedro Marte-Ortiz, Therese A. Rajasekera, Brett R. Loman, Tamar L. Gur, Michael T. Bailey

**Affiliations:** 1Division of Pediatric Gastroenterology, Hepatology and Nutrition, Nationwide Children’s Hospital, Columbus, OH 43205, USA; 2Department of Pediatrics, The Ohio State Wexner Medical Center, Columbus, OH 43210, USA; michael.bailey2@nationwidechildrens.org; 3The Center for Microbial Pathogenesis, The Research Institute, Nationwide Children’s Hospital, Columbus, OH 43205, USA; pedro.marte253@gmail.com (P.M.-O.); brett.loman@nationwidechildrens.org (B.R.L.); 4Oral and Gastrointestinal Microbiology Research Affinity Group, Abigail Wexner Research Institute, Nationwide Children’s Hospital, Columbus, OH 43205, USA; 5Institute for Behavioral Medicine Research, The Ohio State University Wexner Medical Center, Columbus, OH 43210, USA; therese.rajasekera@osumc.edu (T.A.R.); tamar.gur@osumc.edu (T.L.G.); 6Department of Psychiatry & Behavioral Health, The Ohio State University Wexner Medical Center, Columbus, OH 43210, USA

**Keywords:** dextran-sulfate-sodium-induced colitis, IL-17A, social-disruption stressor, anxiety-like behavior, iNOS, network analysis, hippocampal

## Abstract

Stressor exposure increases colonic inflammation. Because inflammation leads to anxiety-like behavior, we tested whether stressor exposure in mice recovering from dextran-sulfate-sodium (DSS)-induced colitis enhances anxiety-like behavior. Mice received 2% DSS for five consecutive days prior to being exposed to a social-disruption (SDR) stressor (or being left undisturbed). After stressor exposure, their behavior was tested and colitis was assessed via histopathology and via inflammatory-cytokine measurement in the serum and colon. Cytokine and chemokine mRNA levels in the colon, mesenteric lymph nodes (MLNs), hippocampus, and amygdala were measured with RT-PCR. SDR increased anxiety-like behaviors, which correlated with serum and hippocampal IL-17A. The stressor also reduced *IL-1β*, *CCL2*, and *iNOS* in the colonic tissue, but increased *iNOS*, *IFNγ*, *IL-17A*, and *TNFα* in the MLNs. A network analysis indicated that reductions in colonic *iNOS* were related to elevated MLN *iNOS* and *IFNγ*. These inflammatory markers were related to serum and hippocampal IL-17A and associated with anxiety-like behavior. Our data suggest that *iNOS* may protect against extra-colonic inflammation, and when suppressed during stress it is associated with elevated MLN *IFNγ*, which may coordinate gut-to-brain inflammation. Our data point to hippocampal *IL-17A* as a key correlate of anxiety-like behavior.

## 1. Introduction

Inflammatory bowel disease (IBD), which includes Crohn’s disease (CD) and ulcerative colitis (UC), is a chronic inflammatory disease involving the gastrointestinal tract. The etiology of IBD is thought to be due to disrupted homeostatic interactions between the mucosal immune system and the commensal microbiota in genetically susceptible individuals [1]. The tissue pathology in these disorders is due to chronic T-cell activation. While UC is often predominated by chronic Th2-cell activation [2], Th1 and Th17 responses also occur [3]. CD, on the other hand, is predominated by chronic Th1/Th17 immune responses [4,5]. Both UC and CD are remitting and relapsing diseases. It is thought that exposure to stressful situations is one key factor that can induce disease relapse [6,7]. Indeed, clinical studies that assessed potential predictors of a relapse in colonic inflammation, such as prior use of non-steroidal anti-inflammatories or antibiotics, prior infections, C-reactive-protein levels, erythrocyte sedimentation rate, and duration of disease have found that exposure to stress is a better predictor of inflammation than all of these other factors [8,9,10]. While many studies have demonstrated that stressor exposure can increase experimental colitis in animal models [11,12], studies have not tested whether stressor exposure can induce a relapse in inflammation. Thus, one purpose of this study was to determine whether exposing mice to a social-defeat stressor would increase colonic inflammation in mice that were recovering from chemical-induced colonic inflammation.

Multiple models have been developed to study certain aspects of human IBD, including pathogen-, T-cell-transfer- and chemical-induced models [11,12,13,14,15,16]. Giving mice dextran sulfate sodium (DSS) in their drinking water induces colonic inflammation that is reflective of human UC [13,14,16]. We have previously shown that challenging mice with the colonic pathogen *Citrobacter rodentium* during exposure to either a prolonged-restraint stressor or to a social-disruption stressor significantly increases colonic histopathology and colonic cytokines and immune factors, such as *iNOS*, *TNFα*, *IFNγ*, and *REG3γ* [11,12]. In addition, others have shown that giving mice DSS to induce colonic inflammation during stressor exposure exacerbates colitis, marked by increases in *IL-6*, *IFNγ*, and *TNFα* and colonic histopathology [17,18]. While these previous studies clearly showed that stress during active colitis increases inflammation, they did not assess the behaviors that are often comorbid in patients with IBD. Gastrointestinal disorders, such as irritable bowel syndrome, have been associated with post-traumatic-stress disorder (PTSD) and anxiety disorders [19], and approximately 25% of IBD patients have comorbid anxiety or depression [20,21]. In patients with IBD, psychological stress has long been reported to increase disease activity, and recent well-designed studies have confirmed that adverse life events, chronic stress, and depression increase the likelihood of relapse in patients with quiescent IBD [22]. However, whether stress exacerbates colitis-associated anxiety and/or depression has not been as well studied. Thus, a second goal of this study was to determine whether exposing mice with chemical-induced colitis to a social-disruption stressor would exacerbate anxiety-like behavior. 

Multiple studies have demonstrated that cytokines are significantly increased in the blood in response to stressful stimuli [23,24]. In mice exposed to stressors that involve repeated social defeat, including the social-disruption (SDR) stressor used in this study, cytokines such as *IL-1* and *IL-6* are increased in the serum, peripheral organs, and the brain and have been shown to have a causative role in anxiety-like behavior [25,26]. *IL-1* and *IL-6* are elevated in patients with IBD [2,4], and in murine models involving colonic inflammation, including DSS-induced colonic inflammation [13,14,16]. Additional cytokines also play a role in colonic inflammation, and *IL-17* has recently been recognized as a key component of intestinal inflammation [5,27,28]. For example, antibodies to IL-17 in the T-cell-transfer model of murine IBD ameliorates inflammation [27]. In addition to its roles in the intestine, IL-17 can also affect the brain and behavior [23,24,29,30]. Both preclinical and clinical studies have shown that *IL-17* levels are increased with stressor exposure and are associated with the development of anxiety and depression [23,24,29,30]. Antibodies that decrease the production of IL-17A have also been shown to decrease anxiety-like behaviors [29]. Thus, mice were exposed to a social-disruption stressor while they were recovering from mild to moderate DSS-induced colitis. Inflammatory mediators and cytokines, including iNOS, IFNγ, IL-1β, IL-6, TNFα, IL-17A, and IL-22 were measured in the colon, mesenteric lymph nodes, blood, and brain regions that are associated with behavioral responses to stress (i.e., hippocampus and amygdala) in order to determine whether they correlated with stressor-induced increases in colonic inflammation and/or anxiety-like behavior. Through the use of network analyses, the data suggest a pathway wherein the stress-induced reduction in protective mucosal immunity is related to the increased inflammatory responses in the mesenteric lymph nodes and serum. These inflammatory responses, particularly the inflammatory cytokine IL-17A, are in turn related to hippocampal IL-17A and anxiety-like behavior. 

## 2. Results

### 2.1. Stressor Exposure Increases Anxiety-like Behavior in Mice Given DSS despite the Lack of Stressor Effects on the Colonic Disease-Activity Index

The administration of 2% DSS for five days did not significantly affect body weight (Figure 1C), but did increase the DAI in both groups. The DAI peaked on day five post administration. This increase in the DAI over the five days occurred in both the control and stressor-exposed groups, with no statistically significant effect of stress on the DAI (Figure 1D). The DAI decreased from day five through day fourteen in all animals, with no differences in stressor-exposed vs. control groups.

To evaluate the effects of stressor exposure on anxiety-like behavior in mice given DSS, we employed two widely used and validated behavioral tests, i.e., light/dark exploration and open-field exploration (Figure 2). Behavior during these tests was recorded approximately 1 h prior to collecting tissues. In the light/dark exploration, stressor-exposed mice spent more time in the dark portion of the box when compared to the control mice (*p* < 0.05, Figure 2A), which is in line with the stressor-induced anxiety-like behavior in previous studies [25,31,32]. There were no differences in the latency to enter the dark portion of the box or in the number of light-dark transitions between stressor-exposed and non-stressed control mice (Figure 2B,C). During open-field exploration, the mice that were exposed to the stressor traveled a shorter distance than the control mice (*p* < 0.05, Figure 2D), but there were no statistically significant differences between the stressor-exposed mice and control mice in the time spent in the periphery, nor in the number of transitions between the center and periphery of the open field (Figure 2E,F). 

### 2.2. Cytokine Expression in the Hippocampus Was Significantly Altered by Stressor Exposure in Mice Given DSS

Cytokine expression in the brain is associated with behavioral responses to stress [29]. Thus, we next tested whether cytokine expression in the hippocampus and amygdala, two brain regions related to anxiety-like behavior in mice, was affected by stressor exposure in mice that were previously given DSS. Of the cytokines and chemokines tested (namely *IL-6*, *IL-17A*, *IL-22*, *CCL2*, *CXCL1*, *CCL25*, and *BDNF*) only the expression of *IL-17A* was significantly increased (*p* < 0.05; Figure 3A) and *TNFα* was significantly decreased (*p* < 0.05; Figure 3B) in the hippocampus. None of the other cytokines and chemokines were significantly different between the control and stressor-exposed mice in the hippocampus (Figure 3C–H) or the amygdala, which had highly variable cytokine gene expression (Table 1). 

### 2.3. Serum Cytokine Levels Were Significantly Affected by Stressor Exposure in Mice Given DSS 

Cytokine and chemokine levels in the serum were assessed, and serum IL-17A, TNFα, and CCL20 were increased in the mice exposed to the social stressor compared to the controls (*p* < 0.05, Figure 4A–C). Stressor exposure had no effect on the levels of IFNγ, IL-10, IL-22, and IL-17E/IL-25 (Figure 4D–G). The remaining cytokines (namely, GM-CSF, IL-2, IL-4, IL-9, IL-13, IL-17F, IL21) were below the detectable limits in the serum.

### 2.4. Stressor-Exposed Mice Given DSS Have Significant Increases in Cytokines in the Mesenteric Lymph Nodes but Few Differences in the Colon

To begin to understand whether increases in serum and brain *IL-17A* may be related to intestinal mucosal immune responses, cytokine and chemokine gene expression was assessed in the mesenteric lymph nodes. The expression of *IL-17A, TNFα, IFNγ,* and *iNOS* was significantly increased in the stressor-exposed group compared to the control group (*p* < 0.05, Figure 5A–D). However, the expression of other cytokines including *IL-1β, IL-6,* and *IL-22* was not significantly different between the two groups (Figure 5E–G). Chemokines *CCL25, CCL2,* and *CXCL1* were evaluated and only the *CCL25* expression was significantly increased in the stressor-exposed mice compared to the control mice (*p* < 0.05, Figure 5H–J). 

To determine whether stressor exposure affected cytokines and chemokines in the colonic tissue of mice given DSS, colonic tissue was collected, and cytokine and chemokine protein and mRNA levels were assessed. Protein was not detectable in any of the samples for many of the analytes, including GM-CSF (detectable in 3/9 controls, 4/9 stress), IL-17E/IL-25 (3/9 controls, 6/9 stress), IFNγ (3/9 controls, 5/9 stress), and IL-22 (3/9 controls, 2/9 stress). In addition, IL-10, IL-13, IL-17F, IL-2, and IL-9 were not detected in any of the colons, but IL-17A, TNFα, CCL20, and IL-21 were detectable in all of the samples. Cytokine protein levels were not significantly affected by stressor exposure (Table 2). However, to further determine if the stressor could affect colonic cytokines and chemokines in mice previously given DSS, the gene expression of several additional cytokines, chemokines, and inflammatory mediators were assessed. Exposure to the stressor significantly reduced the gene expression of IL-1β, iNOS, and CCL2 (Figure 6A,B,I), but had no effect on the expression of any of the other cytokine or chemokine genes (Figure 6C–H,J). Consistent with the lack of increases in colonic cytokines and chemokines, stressor exposure did not affect colonic histopathology (Figure 6K). Representative histologic sections are provided in Figure 6L. 

### 2.5. Neuroinflammation Is Highly Interrelated to Colonic Immune Activation

To understand how gut-brain inflammation is related to anxiety-like behavior, a network analysis was conducted. All of the data collected were tested for all potential Spearman correlations for use in the network analysis. Correlations that were significant at *p* < 0.05 and rho > |0.5| (listed in Appendix A) were plotted in a correlation network for analysis via Cytoscape (Figure 7A). Colonic *iNOS* expression was the most inter-connected factor with a degree (number of correlations) of 17 out of 56 possible relationships. The majority of these relationships (8 of 17) were within the colon, including positive relationships to IL-17A (protein and mRNA) and *CCL2*, which were enhanced by stressor exposure. Notably, colonic *iNOS* expression was also positively related to *TNFα* expression in the hippocampus, and negatively related to *IFNγ* and *CCL25* expression in the MLNs, IL-17A and MIP in circulation, and time spent in the dark. This prominence of colonic *iNOS* in the network was further substantiated by other network parameters, where *iNOS* scored the highest in both Betweenness Centrality (Figure 7B, a measure of network centrality that determines the shortest path (number of connections) between all members of the network and calculates the percentage of paths to which a given member belongs), and Closeness Centrality (Figure 7C, another measure of network centrality similar to Betweenness Centrality, but takes into account the length of each connection; in this case the magnitude of the correlation coefficient).

Multiple markers of immune activation in the MLNs (*TNFα, iNOS*, and *IL-17A* expression) scored a perfect 1 for Clustering Coefficient, meaning that 100% of the parameters they were correlated to were also correlated to one-another (Figure 7D). However, this list of connections was rather short (degrees of two or three), including *iNOS* and *IL-17A* expression being correlated to one-another, and all three MLN markers being related to colon mass. Conversely, *IFNγ* expression in the MLNs had a relatively high Clustering Coefficient score of 0.57 with a degree of 7, indicating a high interconnectivity in this sub-network of stressor-influenced factors that includes colonic *iNOS* and *IL-1β* expression, MLN *CCL25* expression, serum IL-17A and MIP-1α, hippocampal *TNFα* expression, and time spent in the dark. Taken together, these analyses indicated that colonic *iNOS* may be protective against extra-colonic inflammation, and that *IFNγ* in the MLNs may coordinate the gut-brain inflammatory responses that influence anxiety-like behavior as well as systemic IL-17A.

## 3. Discussion

There are extensive bidirectional interactions between the brain and the gut, such that activity in one leads to changes in the other. These interactions are well illustrated in IBDs (either CD or UC) that are marked by excessive intestinal inflammation and commensal microbes. Exposure to stressful situations has been linked to an increased severity of disease symptoms and has been suggested to be a better predictor of a disease flare, which is characterized by abdominal pain, diarrhea, weight loss, or an increase in inflammatory markers such as C-reactive protein, or erythrocyte sedimentation [8,9,10]. In this study, we tested whether exposing mice to the SDR stressor, which involves repeated social defeat and has been shown to exacerbate colonic inflammation, could lead to a flare of inflammation in mice that were recovering from DSS-induced colitis. Contrary to our hypothesis, exposure to the stressor after treatment with DSS did not lead to increased colonic histopathology; the disease activity continued to improve in both stressed and non-stress control mice. However, because IBD patients often have comorbid anxiety and depression [20,21] that are present even when IBD activity is low [33], we also tested behavioral responses to stress when the mice were recovering from colitis. Although the stressor did not exacerbate colitis in this model, it did significantly increase anxiety-like behavior. 

Inflammatory cytokines significantly affect the development of anxiety and depression [23,24,25,29,30] and can be significantly increased during stressful periods [23,34]. In our study, *IL-17A* gene expression was significantly increased in the hippocampus, which is known to contribute to the development of anxiety [35]. IL-17A is well recognized for its effects in neurological disorders, including multiple sclerosis (MS) [36,37,38], Alzheimer’s disease (AD) [39,40], Parkinson’s disease (PD) [41,42,43], and ischemic brain injury [44], and more recently with the development of anxiety and depression [24,29,45]. 

It is not yet known which hippocampal cells express IL-17A. Meningeal γδ T cells have emerged as key sources of IL-17 in the brain, which impacts neuronal activity and subsequent anxiety-like behavior when bound to the IL-17R [29]. We did not collect the meninges in our study, so it is not likely the *IL-17A* gene expression was from meningeal γδ T cells. However, IL-17A is well recognized to affect the blood brain barrier. In vitro, IL-17 can disrupt tight-junction-protein expression in brain endothelial cells, and in vivo, IL-17 can enhance the migration of Th17 cells into the brain [38], making it possible that the observed increase in IL-17A mRNA was due to T cells migrating into the hippocampus. In support of this, others have found Th17 cells in the brains of mice exposed to cumulative mild stress, with increased IL-17A observed in the hippocampus, amygdala, and prefrontal cortex [46]. The cellular sources of stressor-induced IL-17A will be further studied in follow-up experiments, but there is already accumulating evidence that brain IL-17A affects behavioral responses; blocking IL-17 effectively attenuates stressor-induced increases in anxiety- and depressive-like behaviors [29,46]. While this can be due to effects on neurons [29], IL-17 is also known to enhance microglial activation [46,47]. Importantly, murine social stressors involving repeated social defeat are well known to lead to microglial activation, which in turn creates a reactive endothelium in the brain for the recruitment of peripheral monocytes that differentiate into perivascular and parenchymal macrophages [48]. Cytokine production, particularly IL-1β and IL-6 production, by these cells contributes to anxiety-like behavior during social stress [49,50]. The importance of IL-17A has not been studied in this paradigm, but since IL-1β and IL-6 are known to drive IL-17A responses [51], IL-17A warrants further investigation. 

A network analysis was utilized to determine whether anxiety-like behavior and hippocampal *IL-17A* were related to inflammatory responses in the periphery, namely in the colon, mesenteric lymph nodes, and serum. It was originally hypothesized that colonic inflammation would be positively related to anxiety and to hippocampal *IL-17A*, since we have previously shown that stressor exposure enhances colonic inflammation in mice that are challenged with the colonic pathogen *Citrobacter rodentium* [11,12,51,52,53]. However, in this experiment, exposure to the SDR stressor did not increase colonic cytokines; colonic cytokines and inflammatory mediators were either unaffected or significantly reduced by stressor exposure. Although this was surprising, it was consistent with our observation that stressor exposure did not increase colonic histopathology in this model. The lack of stressor-induced increases in colonic cytokines, that have been previously observed, was likely due to the experimental design used in the current experiment. Because IBD is a relapsing/remitting inflammatory disease, we established this model to determine whether stress may induce a relapse of intestinal inflammation. Thus, animals were exposed to the stressor beginning three days after removing DSS, which is a time when colonic inflammation is lower than peak levels. Unlike previous paradigms, this paradigm led to significant reductions in colonic inflammatory mediators, including *IL-1β, CCL2*, and *iNOS*. 

Colonic *IL-1β* and *CCL2* play prominent roles in the immune response to enteric bacteria. For example, signaling through the IL-1 receptor and through the CCL2 receptor (i.e., CCR2) are both necessary for protection against *C. rodentium* infection [54,55]. Despite their importance, colonic *IL-1β* and *CCL2* were not as highly connected in the network analysis as was colonic *iNOS* gene expression. *iNOS* is expressed in normal, healthy colonic tissue [56], but its regulation is important for maintaining homeostasis. The excessive expression of *iNOS* is well recognized in IBD and is thought to be a risk factor for the development of colorectal cancer, due to the effects of reactive nitrogen intermediates (such as nitric oxide and peroxynitrous acid) that damage intestinal tissue [57,58,59]. Stressor exposure in *C. rodentium*-challenged mice leads to the excessive expression of *iNOS*, which is related to tissue damage in this paradigm. However, iNOS also has protective effects in the intestine, where it has direct antimicrobial effects against enteric bacteria [60,61,62] and plays an important role in maintaining the epithelial barrier. The inhibition of NO production has been found to increase intestinal epithelial permeability, whereas NO donors enhance the epithelial barrier. Exposure to stress is recognized to increase epithelial permeability [63], and our results suggest that reduced *iNOS* expression may be involved. Increased epithelial permeability is conducive to bacterial translocation, which is also increased during exposure to stress and is thought to contribute to stressor-induced immune enhancement in the periphery [52,64,65,66]. This is consistent with the current results, where inflammatory responses in the mesenteric lymph nodes and the serum were increased upon stressor exposure. 

Stressor exposure increased the expression of multiple cytokines in the mesenteric lymph nodes, including *IL-17A*, *iNOS*, *TNFα*, and *IFNγ*. Interestingly, the higher expression of *IFNγ* in the mesenteric lymph nodes was inversely related to *IL-1β* and *iNOS* in the colon and positively related to MIP-1α and IL-17A in the serum. The finding that mesenteric-lymph-node *IFNγ* was highly connected in the intestine may not be surprising, since *IFNγ* in the mesenteric lymph nodes plays an important role in the development of aberrant immune responses that occur during IBD and other intestinal inflammatory diseases [67]. However, the finding that mesenteric-lymph-node *IFNγ* was related to anxiety-like behavior was not expected. While this could be a direct relationship to anxiety, *IFNγ* was also related to serum TNFα and IL-17A, the latter of which was also significantly associated with both hippocampal *IL-17A* and to anxiety-like behavior. Thus, the *IFNγ* relationship to anxiety-like behavior may also occur through other cytokines. The cellular source of IFNγ was not determined in the current study, but dendritic cells are important sources of IFNγ, as well as TNFα and iNOS [67]. Because stressor exposure can increase dendritic-cell numbers and activity, it is likely that dendritic cells were increased in the mesenteric lymph nodes of the mice exposed to stress, which is a hypothesis warranting further investigation. The extent to which mesenteric-lymph-node IFNγ can impact the brain and behavior deserves further testing in future studies. 

Serum cytokines appear to be a primary link between mucosal immune responses and behavioral responses during stress. Mesenteric-lymph-node *IFNγ* was related to serum IL-17A and TNFα. Interestingly, other serum cytokines, including IFNγ, IL-10, IL-22, or IL-17E/IL-25 were unaffected by stressor exposure. Multiple studies have linked TNFα with anxiety-like behavior in other animal models [68,69] and human studies [70,71], but less is known about serum IL-17A and anxiety-like behavior. However, given the effects that IL-17A can have on the blood brain barrier, neurons and microglia, its demonstrated importance in neurological disorders, and its importance in intestinal inflammation and homeostasis, we propose that serum IL-17A is an important cytokine that can link intestinal immunity to behavioral responses, particularly during stressful periods. Patients with inflammatory diseases such as IBD often have comorbid anxiety and exposure to stress often exacerbates the disease severity [8,72,73,74]. Despite this realization, the mechanisms by which this occurs are not well understood. Our findings have led us to propose a paradigm wherein stressor exposure during periods of mild colitis leads to an inhibition of colonic cytokines and inflammatory factors (such as IL-1β and iNOS) that have protective responses to intestinal bacteria. A reduction in these defenses creates an environment that allows bacteria, or bacterial products, to translocate to the interior of the body where they stimulate inflammatory responses and manifest as increases in cytokines in the mesenteric lymph nodes and serum, including IFNγ, IL-17A, and TNFα. Based on network analyses, we suggest that these cytokines play a key role in linking immune responses in the colon, mesenteric lymph nodes, and serum to hippocampal IL-17A and subsequent anxiety-like behavior. This suggests that inhibiting IL-17A may improve comorbid anxiety in patients with IBD. However, development of anti-IL-17 therapies (e.g., Secukinumab and Brodalumab) were halted due to worsening IBD severity during clinical trials [75]. Alternative strategies to decrease IL-17 are currently being developed, such as antibodies of the IL-23 receptor that is known to drive IL-17 responses [75]. As these treatments become available, it will be interesting to test whether they can inhibit anxiety-like behavior. 

## 4. Materials and Methods

### 4.1. Animals

Male C57BL/6 mice were purchased from Charles River Laboratories at 6–8 weeks of age. Mice were maintained at the animal resources core facilities in a controlled room (temperature 20–25 °C, humidity 70–73%, and 12-h light-dark cycle) and housed 3 per cage with free access to standard food and water. Mice were acclimated to these conditions one week prior to the start of the study. All experiments were approved by the Institutional Animal Care and Use Committee at The Research Institute at Nationwide Children’s Hospital (Columbus, OH, USA) protocol number AR16-00059. 

### 4.2. Experimental Design

Multiple studies have shown that stressor exposure prior to or during DSS worsens the degree of colitis [17,18]. However, this experiment was designed to evaluate whether exposure to a stressor during the resolution of DSS-induced colitis could reactivate and enhance colonic inflammation. 

We conducted a single experiment that included a total of 18 male mice 7–9 weeks old that received 2% DSS (MW ca 40,000; Alfa Aesar^TM^, (Ward Hill, MA, USA)) from day 1–5 in order to cause a mild/moderate degree of colonic inflammation. Proper concentrations of DSS were maintained by changing the drinking water every three days. Regular drinking water was placed in all cages on day 5 and for the remainder of the experiment. Half of the mice (*n* = 9) were exposed to a social stressor, i.e., social disruption (SDR), from days 9–13. SDR is a well-studied social stressor [12,76,77] that involved, in this study, placing an aggressive male (retired C57BL/6 male breeder) into the cage of 3 resident mice so that the aggressor could defeat the resident mice over a 2-h period. At the end of the 2-h period, the aggressors were removed and the residents were left undisturbed until the following day. This paradigm was repeated for 5 consecutive days. In addition to mice exposed to the SDR stress, half of the mice (*n* = 9) were not exposed to a stressor and served as controls. On the morning of day 14, behaviors in the light/dark-exploration and open-field-exploration tests were observed for all of the mice prior to collecting colonic tissue, mesenteric lymph nodes, serum and the brain (Figure 1A,B). Female mice were not used for this experiment because female mice are not aggressive, and thus are not defeated during the SDR paradigm. In our experience, we have observed that DSS has similar effects on mice within a single cage and that there is variability between cages. Therefore, we utilized 3 replicate cages of mice in our experiments. 

The mice were monitored daily for water/DSS consumption, weight, stool consistency, and presence of blood in the stool throughout the experiment. Change in body weight was calculated as the percent difference between the body weight on day 0 and the body weight on each individual day. Loss of body weight was scored on a scale from 0–4 (<1% = 0, 1–5% = 1, 5–10% = 2, 10–15% = 3, ≥15% = 4). Stool consistency was scored on a scale from 0–3, where a score of 0 indicated normal stool, 1 indicated soft-formed stool, 2 indicated mushy stool without a pellet shape, and 3 indicated watery stool. Presence of blood was also scored on a scale from 0–3, with a score of 0 indicating no visible blood, 1 indicating a speck of blood, 2 indicating blood mixed with stool, and 3 indicating very bloody stool with rectal bleeding. The disease-activity index (DAI) was calculated as the total of the sums of the scores of body-weight loss, stool consistency, and presence of blood, resulting in a score range from 0 (unaffected) to 10 (severe colitis) [34]. 

### 4.3. Behavioral Tests

#### 4.3.1. Light/Dark Exploration

The light/dark-exploration apparatus consisted of an acrylic box measuring 60 × 45 × 30 cm and was divided into two compartments by a black plexiglass wall with an opening that allowed mice to move freely between the two compartments. The light compartment occupied two thirds of the box and was illuminated by a 13-watt white bulb with illuminance at the center of the chamber being ~500 lux; the dark compartment was enclosed by black plexiglass. Mice were placed in the center of the illuminated compartment and allowed to explore for 300 s. Animal movement was recorded using a video camera. Behavior was scored from the videos. Latency to the first entrance from the light to the dark compartment (defined as all 4 paws inside the compartment), the amount of time spent in the dark and in the light portions of the box, and the number of transitions between the light and the dark portions of the box were recorded. 

#### 4.3.2. Open Field Exploration

The open-field apparatus was composed of a clear acrylic enclosure measuring 40 × 40 × 38 cm. Mice were placed in the front of the left corner and allowed to explore for 300 s. Behavior was scored using 16 × 16 photo-beam configuration analyzed by the software Photo-beam Activity System version 2 (San Diego Instrument’s, San Diego, CA, USA) to track the animal path within the enclosure. The amount of time the mice spent in the center of the open field (defined as the inner 30 × 30 cm) or periphery (defined as the outer 10 × 10 cm) was recorded, along with the number of transitions between the center and periphery of the open field, and the total distance were recorded. 

### 4.4. Histopathology

The distal colon was taken for blinded histopathology scoring using a validated scoring system [78]. A score of 0 represented no inflammation; 1—mild inflammation is depicted with one or a few multifocal mononuclear cell infiltrates in the lamina propria accompanied by minimal epithelial hyperplasia and slight to no depletion of mucus from goblet cells; 2—moderate inflammation is depicted with several multifocal cell infiltrates in the lamina propria, mild epithelial hyperplasia, mucin depletion, and small epithelial erosions; 3—moderate/severe inflammation is depicted with inflammation involving the submucosa, crypt abscesses, moderate epithelial hyperplasia with mucin depletion.; and 4—severe inflammation with lamina propria infiltration, architectural distortion, crypt abscesses, and ulcers. 

### 4.5. Semiquantitative Real-Time PCR

Total RNA was isolated from the distal colon, mesenteric lymph nodes (MLNs) hippocampus, and amygdala using Tri-zol reagent (Invitrogen, Carlsbad, CA, USA) following manufacturer protocols and subjected to double lithium-chloride (2.5 M) purification for the removal of DSS in the distal colon and MLNs [79]. All mouse mRNA primers were purchased from (Integrated DNA Technologies, Redwood city, CA, USA). Real-time PCR was carried out with a QuantStudio 3 system (Applied Biosystems, Bedford, MA, USA). The housekeeping gene Eef2 was used, and the relative amount of transcript was determined using the comparative cycle threshold (C_t_) method as previously described [52,80]. Primer sequences are provided in Appendix A. Data were expressed as a fold change from the control group. 

### 4.6. Serum Measurement of Cytokines

Serum and distal colonic cytokines were measured using U-plex T cell combo kit (GM-CSF, IFNγ, IL-2, IL-4, IL-9, IL-10, IL-13, IL-17A, IL-17E/IL-25, IL-17F, IL-21, IL-22, MIP-3α (a.k.a CCL20), TNFα) from Meso Scale Diagnostics (Rockville, MD, USA) per manufacturer’s instructions. The plates were analyzed on the MESO QuickPlex SQ 120 imager. Calculation of cytokine concentrations was subsequently determined by 4-parameter logistic non-linear regression analysis of the standard curve. Cytokine levels were expressed as pg/mL for serum and were adjusted to pg/mg of colon tissue weight where appropriate.

### 4.7. Statistical Analysis

A mixed-factor ANOVA was performed for DAI with the day of analysis as a repeated measure and the group as a between measure. Normality testing was performed to determine if the data were normally distributed. Data were considered normally distributed if the Shapiro–Wilk test *p*-value was >0.05. Normally distributed data were analyzed using a t-test and not normally distributed data were analyzed using a Mann–Whitney test. All collected data were tested for all potential Spearman correlations for use in network analyses. For network and correlation analyses, gene-expression data were expressed as 1/ΔCt and protein data were expressed as pg/mL. Correlations that were significant at *p* < 0.05 and rho > |0.5| were plotted in a correlation network for analysis via Cytoscape version 3.9.1 (Cytoscape Consortium, San Diego, CA, USA), where the absolute value of correlation coefficients were plotted as edges, and parameters were represented as nodes. The Betweenness Centrality, Closeness Centrality, and Clustering Coefficient were calculated in Cytoscape. Betweenness Centrality calculates the shortest, unweighted path (number of edges) between all pairs of nodes (excluding the node in question), then divides the number of shortest paths on which a node lies by the total number of shortest paths in the network. Higher values indicate more influence on the flow of information through the network, with a value of 1 indicating that a given node lies on all the shortest paths between all other nodes. Closeness Centrality calculates the reciprocal of the average, unweighted, shortest path between the node in question and all other nodes. In this case, higher values indicate more direct relationships with other variables, with a value of 1 indicating that a given node is directly related to all other nodes. Clustering Coefficient is calculated as the number of triangles (loop of three edges with immediately neighboring nodes) that pass through the node in question divided by the total possible number of triangles. Higher values indicate that a node and its neighbors are highly interrelated, with a value of 1 indicating that all neighbors of a given node are also directly related (in this case, all neighbors are also correlated to one another). All other data were analyzed using SPSS statistical software version 26 (IBM Corp, Armonk, NY, USA).

## Figures and Tables

**Figure 1 ijms-23-02000-f001:**
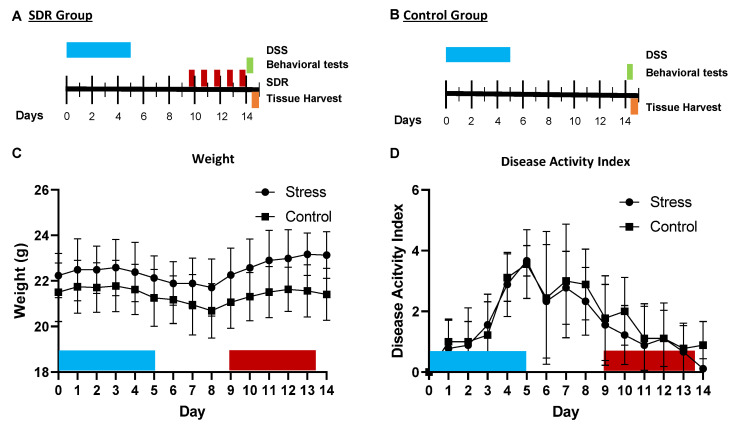
Experimental Design. (**A**,**B**) Depiction of the experimental design for the stressor-exposed group and control group. (**C**) Weight of the course of the experiment for both groups. (**D**) Disease-activity index was calculated as the total of the sum of body-weight loss, stool consistency, and presence of blood resulting in a score ranging from 0 (unaffected) to 10 (severe colitis).

**Figure 2 ijms-23-02000-f002:**
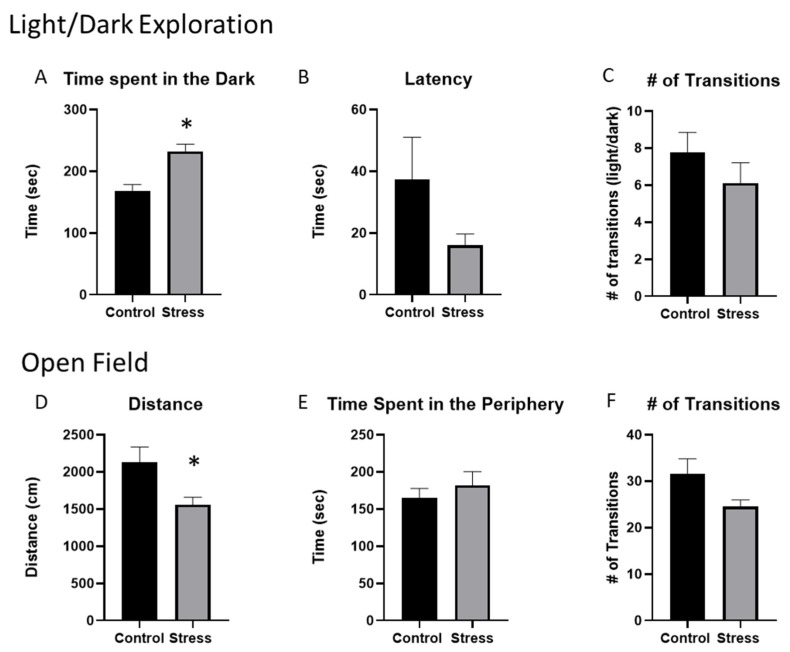
Exposure to the social-disruption stressor increases anxiety-like behavior in mice previously given DSS. Mice were treated with 2% DSS for five days. Four days after ending DSS treatment, mice were either exposed to the SDR stressor for six days or were left undisturbed as a control. Behavior was observed to assess anxiety-like behavior in the light/dark-preference test or in the open field. (**A**) Time spent in the dark was significantly higher in the stress group vs. the control group * *p* < 0.05. (**B**,**C**) Latency to enter the dark portion of the box and the number of transitions between the light and the dark portions of the box were not different between stressor-exposed and non-stressed control mice. (**D**) The average distance traveled in the open field was significantly lower in mice exposed to the stressor compared to the non-stress control group. * *p* < 0.05. (**E**) The amount of time spent in the periphery of the open field and (**F**) the number of transitions between the center and the periphery of the open field were not significantly different between stressor-exposed and control mice.

**Figure 3 ijms-23-02000-f003:**
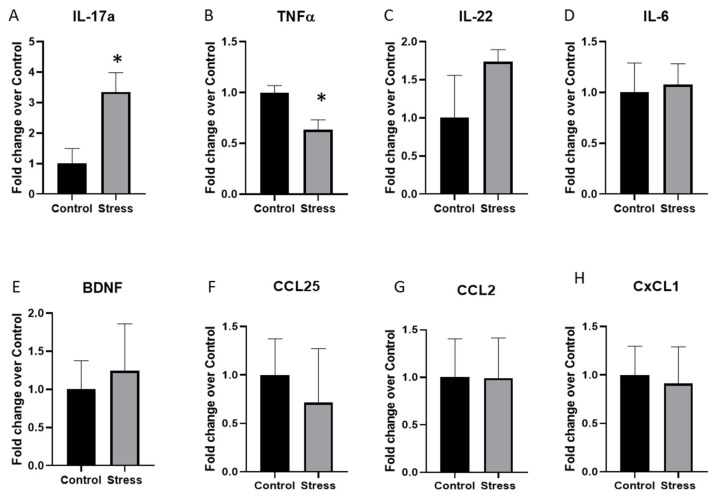
Exposure to the social-disruption stressor changes hippocampal cytokine and chemokine gene expression in mice previously given DSS. After behavioral testing, mice were euthanized and cytokine gene expression in the hippocampus was assessed using real-time PCR. (**A**) *IL-17A* gene expression in the hippocampus was significantly higher in the stress group vs. the control group * *p* < 0.05. (**B**) *TNFα* gene expression in the hippocampus was significantly higher in the stress group vs. the control group * *p* < 0.05. (**C**–**H**) *IL-22, IL-6, BDNF, CCL25, CCL2*, and *CXCL1* gene expression was not different between the stress and the control groups. Gene expression was calculated using the delta-Ct method and is expressed as a fold change of the control group.

**Figure 4 ijms-23-02000-f004:**
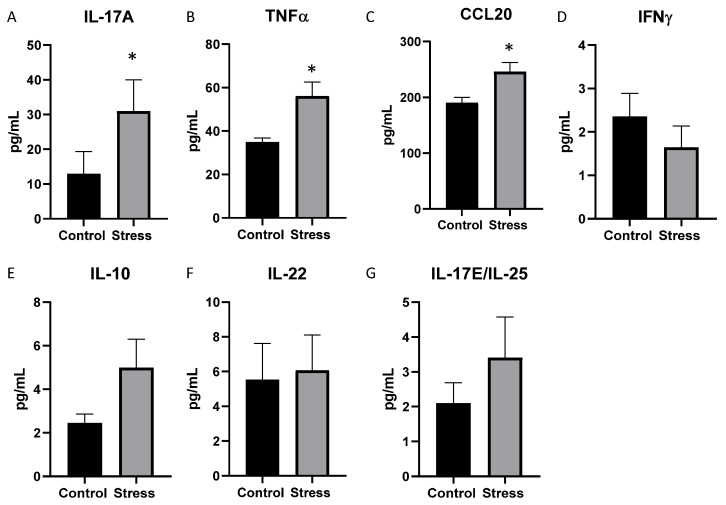
Cytokine levels in the serum are significantly affected by stressor exposure in mice previously given DSS. Serum cytokine levels were measured after testing behavior in mice previously treated with DSS. Serum levels of (**A**) IL-17A, (**B**) TNFα, and (**C**) CCL20 were significantly increased in mice exposed to the stressor compared to non-stress control mice. * *p* < 0.05. In contrast, serum levels of (**D**) IFNγ, (**E**) IL-10, (**F**) IL-22, and (**G**) IL-17E/IL-25 were not significantly different between stressor-exposed and non-stressed control mice.

**Figure 5 ijms-23-02000-f005:**
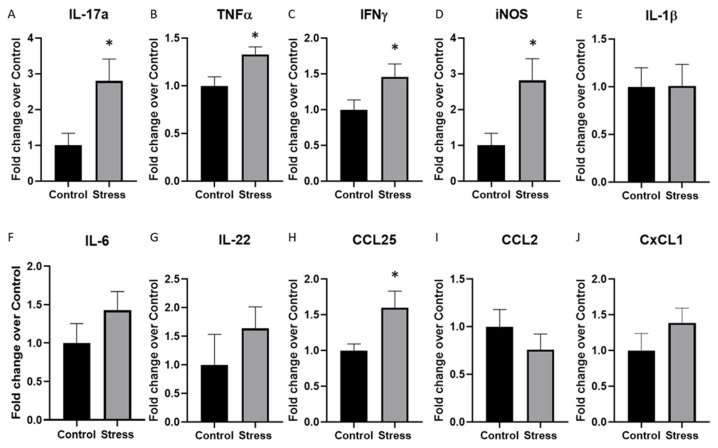
Cytokine and chemokine gene expression is significantly increased in the mesenteric lymph nodes of stressor-exposed mice previously treated with DSS. Cytokine and chemokine gene expression was measured in the mesenteric lymph nodes using real-time PCR. The expression of (**A**) *IL-17A*, (**B**) *TNFα*, (**C**) *IFNγ*, and (**D**) *iNOS* was significantly higher in mice exposed to the stressor compared to control mice. However, there were no differences in (**E**) *IL-1β*, (**F**) *IL-6*, or (**G**) *IL-22*. The expression of (**H**) *CCL25* was significantly higher in stressor-exposed vs. non-stressed control mice, but there were no differences in the mesenteric-lymph-node expression of (**I**) *CCL2* or (**J**) *CXCL1*. * *p* < 0.05 stress vs. control.

**Figure 6 ijms-23-02000-f006:**
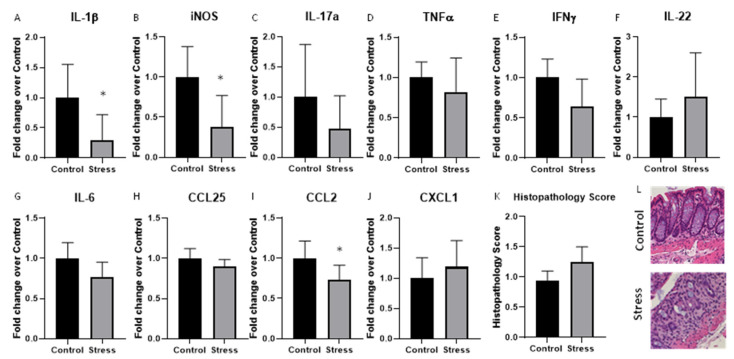
Colonic cytokine gene expression is significantly reduced by stressor exposure in mice previously given DSS. Colonic cytokine and chemokine gene expression was measured in mice previously given DSS using real-time PCR. The expression of (**A**) *IL-1β* and (**B**) *iNOS* in the distal colon was significantly decreased in the stress group vs. the non-stress controls * *p* < 0.05. In contrast, (**C**) *IL-17A*, (**D**) *TNFα*, (**E**) *IFNγ* (**F**) *IL-22*, and (**G**) *IL-6* were not different in stressor-exposed vs. non-stress control mice. The chemokines (**H**) *CCL25* and (**J**) *CXCL1* were the same in stress and control mice, but (**I**) *CCL2* was significantly lower in stress vs. control mice. * indicates *p* < 0.05 for stress vs. control mice. (**K**) Colonic histopathology scores were similar in control mice and mice exposed to stress after being previously treated with DSS. (**L**) Representative images of a control and stressor exposed mice. 20× Magnification.

**Figure 7 ijms-23-02000-f007:**
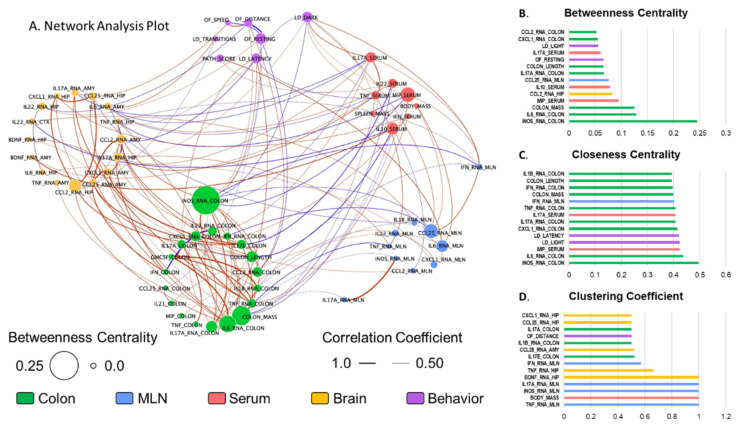
Stressor-induced changes in colonic, mesenteric-lymph-node, serum and brain cytokines are related to each other and to anxiety-like behavior. (**A**) Data were tested using Spearman correlations for use in network analyses. Correlations that were significant at *p* < 0.05 and rho > |0.5| were plotted in a correlation network using Cytoscape. Nodes are colored based on body region and scaled based on Betweenness Centrality. Edges are colored blue for negative correlations and red for positive correlations; edge thickness is scaled based on the absolute value of the correlation coefficient. *iNOS* expression in the colon, time spent in the dark, IL-17 in the serum, and *IFNγ* and *IL-17A* expression in the mesenteric lymph nodes are pulled out of their respective tissue clusters to highlight their relationships. The top 14 values for each network parameter calculated in Cytoscape, including (**B**) betweenness centrality, (**C**) closeness centrality, and (**D**) clustering coefficient are shown.

**Table 1 ijms-23-02000-t001:** Amygdala cytokine and chemokine expression.

	Control	Stress	*p* Value
*IL-17a*	1.0 + 1.36	0.52 + 1.6	0.66
*TNFα*	1.0 ± 0.13	0.9 ± 0.21	0.51
*IL-22*	1.0 ± 0.91	0.59 ± 0.47	0.52
*IL-6*	1.0 ± 0.14	0.9 ± 0.36	0.67
*BDNF*	1.0 ± 0.16	1.46 ± 0.47	0.27
*CCL25*	1.0 ± 0.59	0.88 ± 0.55	0.817
*CCL2*	1.0 ± 0.77	1.49 ± 0.38	0.52
*CxCL1*	1.0 ± 0.61	0.75 ± 0.66	0.65

Gene Expression: Fold change over control. Data are mean ± std error. *t*-tests were only conducted with positive samples.

**Table 2 ijms-23-02000-t002:** Colon cytokine levels.

	Control	Stress	*p* Value
IL-17A	0.86 ± 0.28	0.49 ± 0.17	0.29
TNFα	14.50 ± 5.76	10.35 ± 2.16	0.51
CCL20	27.16 ± 8.24	32.71 ± 6.99	0.62
IL-21	2.70 ± 1.09	1.27 ± 0.32	0.26
IL-22	0.72 ± 0.33 (3)	0.41 ± 0.44 (2)	0.54
IL-17E/IL-25	0.18 ± 0.05 (3)	0.10 ± 0.03 (6)	0.20
IFNγ	0.86 ± 0.41 (3)	0.39 ± 0.13 (5)	0.23
GM-CSF	0.94 ± 0.47 (3)	0.06 ± 0.02 (4)	0.07

Colon Protein: pg/mg tissue. Number in () indicates number of samples with detectable cytokine levels. Data are mean ± std error unless only two samples had detectable protein levels (in which case data are mean ± range). *t*-tests were only conducted with positive samples.

## Data Availability

Data are available upon request from the corresponding author.

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
