# Peer review of "Stressor-Induced Increases in Circulating, but Not Colonic, Cytokines Are Related to Anxiety-like Behavior and Hippocampal Inflammation in a Murine Colitis Model"

_ijms, 2022, doi:10.3390/ijms23042000_

Round 1

Reviewer 1 Report

The manuscript is generally well-written. Please see below for my specific comments.

Specific comments:

  1. As per the journal's guidelines, the abstract should be a total of about 200 words maximum.
  2. "While these previous studies clearly showed that stress during active colitis increases inflammation, they did not assess behaviors that are often comorbid in patients with IBD. Approximately 25% of IBD patients have comorbid anxiety or depression,[19, 20]" - gut inflammation and the 'leaky gut' response have also been implicated in the pathogenesis of irritable bowel syndrome (IBS) (citation: pubmed.ncbi.nlm.nih.gov/30288077). IBS has in turn been associated with PTSD and anxiety disorders (citation: pubmed.ncbi.nlm.nih.gov/30144372).
  3. "... whether stress exacerbates these disorders" - this is not true. Psychological stress has long been reported to increase disease activity in inflammatory bowel disease (IBD), and recent well-designed studies have confirmed that adverse life events, chronic stress, and depression increase the likelihood of relapse in patients with quiescent IBD (citation: ncbi.nlm.nih.gov/pmc/articles/PMC6821654).
  4. The nature of replication in the experimental design is unclear, and the assessment of uncertainty in the reported measurement is absent or unclear.
  5. "A total of 18 male mice" - at what age?
  6. A clinical perspective is lacking in the discussion section of this paper. Would the findings translate clinically?
  7. In recent years, the combinations of gas chromatography-quadrupole time of flight mass spectrometry (GC-Q-TOF/MS) and liquid chromatography-quadrupole time of flight mass spectrometry (LC-Q-TOF/MS) has been applied successfully in numerous metabolomics studies to achieve more sensitive and accurate metabolic profiling and mechanistic understandings (citation: pubmed.ncbi.nlm.nih.gov/30056340). This is an area for future study the authors could propose.
  8. "... was funded by the Nationwide Children’s Hospital Foundation" - if available, please provide the actual funding/grant number.
  9. Please provide a data availability statement.
  10. There are no materials provided under Appendix A and B.

Author Response

Reviewer 1 Specific comments:

  1. As per the journal's guidelines, the abstract should be a total of about 200 words maximum.

Response:  Thank you, the abstract was adjusted.

  1. "While these previous studies clearly showed that stress during active colitis increases inflammation, they did not assess behaviors that are often comorbid in patients with IBD. Approximately 25% of IBD patients have comorbid anxiety or depression,[19, 20]" - gut inflammation and the 'leaky gut' response have also been implicated in the pathogenesis of irritable bowel syndrome (IBS) (citation: pubmed.ncbi.nlm.nih.gov/30288077). IBS has in turn been associated with PTSD and anxiety disorders (citation: pubmed.ncbi.nlm.nih.gov/30144372).

Response:  We have included language (lines 69-73) acknowledging that gastrointestinal disorders, such as irritable bowel syndrome, have long been associated with PTSD and anxiety. 

  1. "... whether stress exacerbates these disorders" - this is not true. Psychological stress has long been reported to increase disease activity in inflammatory bowel disease (IBD), and recent well-designed studies have confirmed that adverse life events, chronic stress, and depression increase the likelihood of relapse in patients with quiescent IBD (citation: ncbi.nlm.nih.gov/pmc/articles/PMC6821654).

Response:  We agree with the reviewer that psychological stress has been reported to increase disease activity and to increase disease relapse.  While this is a topic that we are interested in, the purpose of this study was to determine whether stress also increases colitis-associated anxiety.  We have clarified this on lines 73-78.

  1. The nature of replication in the experimental design is unclear, and the assessment of uncertainty in the reported measurement is absent or unclear.

Response: We have clarified that the data are from a single experiment containing a sample size of n=9 mice per group. This now emphasized on lines 423, 427-428, and 439-441.

  1. "A total of 18 male mice" - at what age?

Response:  Mice were purchased at 6-8 weeks old and acclimated for 1 week. Mice were 7-9 weeks old at the initiation of the experiment. This is now on line 411.

  1. A clinical perspective is lacking in the discussion section of this paper. Would the findings translate clinically?

Response:  Thank you for asking this question.  We design all of our studies with translation in mind, but the translational component is not always straight forward, which was the case with this study.  However, we do think that serum (and brain) IL-17A links intestinal inflammation to anxiety and depression, and strategies to reduce IL-17A may improve mental health in patients with IBD.  This is now discussed on lines 386-407.  

  1. In recent years, the combinations of gas chromatography-quadrupole time of flight mass spectrometry (GC-Q-TOF/MS) and liquid chromatography-quadrupole time of flight mass spectrometry (LC-Q-TOF/MS) has been applied successfully in numerous metabolomics studies to achieve more sensitive and accurate metabolic profiling and mechanistic understandings (citation: pubmed.ncbi.nlm.nih.gov/30056340). This is an area for future study the authors could propose.

Response:  We agree with the reviewer that this is a potential area for future studies, and in fact we have begun using this methodology for our studies involving the microbiome and intestinal physiology.  We hope to apply this technique to our studies of colitis in the near future.

  1. "... was funded by the Nationwide Children’s Hospital Foundation" - if available, please provide the actual funding/grant number.

Response: The actual funding does not have a grant number.

  1. Please provide a data availability statement.

Response: We now include a statement that data are available upon request. 

  1. There are no materials provided under Appendix A and B.

Response:  Thank you for pointing this out.  We have deleted Appendix B, but have included Supplementary Tables for Appendix A.

Reviewer 2 Report

In the present work the researchers investigated association between stressor exposure and anxiety-like behavior in mice with DSS-driven colitis and, measuring expression of key cyto- and chemokines on both mRNA and protein levels in different tissues, tried to reconstruct molecular mechanism of this interconnection. It was found that aggressive male mice-induced stress did not affect the severity of colitis, however, stimulated anxiety-like behavior in colitis-bearing mice and significantly modulated expression of some cyto/chemokines in their hippocampus, serum, mesenteric lymph nodes and colon tissue. Finally, using network analysis, authors concluded that neuroinflammation is highly correlated with colonic immune activation, iNOS can play protective effect against extra-colonic inflammation and IFN-γ, IL-17A, and TNF-α play a key role in linking immune responses in mentioned tissues as well as anxiety-like behavior. Undoubtedly, the research is highly interesting; however, in the current form it is not suitable for publication in IJMS and should be rejected.

Analysis of the manuscript revealed a range of major and minor comments:

Major comments:

  1. It is completely not clear how final network depicted in Fig. 7 was created? If you reconstruct your network based on Spearman correlation coefficients, could you please explain, for instance:

(a) Why was IL1B_mRNA_colon, being down-regulated in stressor exposed colitis-induced mice (Fig. 6), correlate with GMCSF, which was undetectable in colonic tissue (please, see lines 186-187)?!

(b) In Fig. 7 in lower green module IL17E_colon (also known as IL-25) can be found. Is it correct? However, you did not measure its expression in colon tissue by RT-PCR or cytoplex analysis. Moreover, U-plex T cell combo kit used to measure protein levels did not contains antibodies against IL17E (according to data presented in Section 4.6). If it is misprint and IL17E should be consider as IL17F (undetectable in colon tissue (line 187)), why was this protein correlate with IL22_serum, CXCL1_RNA_AMY, CCL25_RNA_AMY and CCL2_RNA_AMY, however other undetectable proteins, such as GMCSF, IL10, IL13 (line 187) did not correlate with them?

(c) Why was down-regulated iNOS_RNA_colon correlate with unchanged CCL2_RNA_AMY?

(d) And a lot of similar questions about this network.

If you use reconstructed network to form some conclusions and hypothesis, the description of methods of network analysis have to be more informative and the network should be understandable.

  1. In my opinion, the diagram demonstrated weight loss of control and stressed groups during the experiment as well as representative photos of inflamed colon tissues in both mentioned groups should be added in the manuscript because disease activity index is mostly subjective. Moreover, authors did not explain how histopathology samples were prepared and based on which characteristics mild and moderate inflammatory changes in colon tissues were discriminated from each other. Please, correct.

Minor comments:

  1. 1c. Please, correct the legend.
  2. Line 115. Please, write “h” instead of “hr”.
  3. lines 116, 120, 129, 132, 150, 151, 165, 173, 183, 203, 205. p<.05 is incorrect. Please, correct (p<0.05).
  4. Sections Introduction and Discussion. According to author’s guideline of IJMS, references should be placed after word but before punctuation marks. Please, correct.
  5. Table 1. Could you explain, please, why standard deviation values in Amygdala IL-17a are so high (more than average value). Is it correct? Can we trust such data?
  6. 5, Fig. 6. Please, correct “CxCL1”.
  7. line 187 please, delete superfluous "including".
  8. line 187 please, correct INF-γ.
  9. How can you explain the absence of protein expression of IFN-γ in colon tissue (line 187), whereas its mRNA level was detectable by RT-PCR?
  10. Table 2. Please, use either MIP-1α or CCL3, but not mix of them.
  11. Section 4.5. The sequences of all PCR-primers used in the study should be added.
  12. If you use cytoplex assay to measure protein levels of cyto/chemokines, why was a part of them but not all panel depicted in Fig. 4 ?

Author Response

Reviewer 2 Comments and Suggestions for Authors

In the present work the researchers investigated association between stressor exposure and anxiety-like behavior in mice with DSS-driven colitis and, measuring expression of key cyto- and chemokines on both mRNA and protein levels in different tissues, tried to reconstruct molecular mechanism of this interconnection. It was found that aggressive male mice-induced stress did not affect the severity of colitis, however, stimulated anxiety-like behavior in colitis-bearing mice and significantly modulated expression of some cyto/chemokines in their hippocampus, serum, mesenteric lymph nodes and colon tissue. Finally, using network analysis, authors concluded that neuroinflammation is highly correlated with colonic immune activation, iNOS can play protective effect against extra-colonic inflammation and IFN-γ, IL-17A, and TNF-α play a key role in linking immune responses in mentioned tissues as well as anxiety-like behavior. Undoubtedly, the research is highly interesting; however, in the current form it is not suitable for publication in IJMS and should be rejected.

Analysis of the manuscript revealed a range of major and minor comments:

Major comments:

  1. It is completely not clear how final network depicted in Fig. 7 was created?

Response:  We agree that this was not clear enough in the original submission.  We have taken additional steps to clarify how the network analysis was completed and visualized. Figure 7 was created using the program Cytoscape, as detailed in section 4.7 of the Materials and Methods: “All data collected were tested for all potential Spearman correlations for use in network analyses. Correlations that were significant at p<0.05 and rho>|0.5| were plotted in a correlation network for analysis via Cytoscape, where correlation coefficients are represented as edges, and parameters are represented as nodes. Betweenness centrality, closeness centrality, and clustering coefficient were calculated in Cytoscape. For network and correlation analyses, gene expression data was expressed as 1/ΔCt and protein data was expressed as pg/ml.”

… as well as the figure 7 legend: “(A) Data were tested using Spearman correlations for use in network analyses. Correlations that were significant at p < 0.05 and rho>|0.5| were plotted in a correlation network using Cytoscape. Nodes are colored based on body region and scaled based on Betweenness Centrality. Edges are colored blue for negative correlations and red for positive correlations; edge thickness is scaled based on the correlation coefficient. iNOS expression in the colon, time spent in the dark, IL17 in the serum, and IFN and IL17A expression in the mesenteric lymph nodes are pulled out of their respective tissue clusters to highlight their relationships. The top 14 values for each network parameter calculated in Cytoscape, including (B) betweenness centrality, (C) closeness centrality, and (D) clustering coefficient are shown.”

  1. If you reconstruct your network based on Spearman correlation coefficients, could you please explain, for instance:

(a) Why was IL1B_mRNA_colon, being down-regulated in stressor exposed colitis-induced mice (Fig. 6), correlate with GMCSF, which was undetectable in colonic tissue (please, see lines 207-209)?!

Response:  We apologize that this was confusing. Several cytokines were detectable in some of the samples, but not all of the samples.  These were inadvertently reported as not-detectable in the text, but samples with detectable levels were used in the network analysis.  We have revised the text to more accurately state that GM-CSF was detectable in 3/9 samples from control mice and 4/9 samples from stressor-exposed mice.  Means and standard errors are also now found in Table 2 along with appropriate notation for the proportion of samples with undetectable amounts.  This was also true for IFN-γ, IL-22, and IL17E/IL-25.  Every sample with detectable cytokines was used in the network analysis.  This is now clarified on line 510-511. 

(b) In Fig. 7 in lower green module IL17E_colon (also known as IL-25) can be found. Is it correct? However, you did not measure its expression in colon tissue by RT-PCR or cytoplex analysis. Moreover, U-plex T cell combo kit used to measure protein levels did not contains antibodies against IL17E (according to data presented in Section 4.6). If it is misprint and IL17E should be consider as IL17F (undetectable in colon tissue (line 187)), why was this protein correlate with IL22_serum, CXCL1_RNA_AMY, CCL25_RNA_AMY and CCL2_RNA_AMY, however other undetectable proteins, such as GMCSF, IL10, IL13 (line 187) did not correlate with them?

Response:  We have clarified which cytokines are measured by the U-Plex T cell Combo Kit on lines 497-499 (which can also be found here: http://www.mesoscale.com/en/products/u-plex-t-cell-combo-mouse-k15098k/.  This kit does assess IL-17E, which was detectable in some, but not all samples.  We have clarified this in response (a) above, which should also address this point here.

(c) Why was down-regulated iNOS_RNA_colon correlate with unchanged CCL2_RNA_AMY?

Response:  We apologize that this was confusing.  Unfortunately, with so many significant correlations, the network became cluttered and it was not possible to have any overlap. The line that appears to be connecting iNOS_RNA_colon to CCL2_RNA_AMY is actually connecting iNOS_RNA_colon to TNF_RNA_HIP. Since we could not design the network without any overlap, we now include the actual correlation coefficients as Supplemental Table 2 and point the reader to these correlations (lines 239-241).

(d) And a lot of similar questions about this network.

If you use reconstructed network to form some conclusions and hypothesis, the description of methods of network analysis have to be more informative and the network should be understandable.

Response:   In an effort to guide readers that may be less familiar with canonical network analyses, we have described each of the three major methods utilized in this paper in section 4.7 of the Materials and Methods.  We hope the reviewer agrees that along with the inclusion of the Spearman correlation coefficients, this new information helps with the interpretation of the network analysis.

In my opinion, the diagram demonstrated weight loss of control and stressed groups during the experiment as well as representative photos of inflamed colon tissues in both mentioned groups should be added in the manuscript because disease activity index is mostly subjective. Moreover, authors did not explain how histopathology samples were prepared and based on which characteristics mild and moderate inflammatory changes in colon tissues were discriminated from each other. Please, correct.

Response:  We regret that this was not made clear in the initial submission.  We now include a description of how colonic tissue was processed and scored (section 4.4) and have included representative images from Control and Stress groups (Fig. 6L) to compliment the histopathology score found in Fig. 6K.  We have also included weight loss changes in Figure 1C.

Minor comments:

  1. 1c. Please, correct the legend.

Response: Thank you the legend has been corrected.

  1. Line 115. Please, write “h” instead of “hr”.

Response: Correction made

  1. lines 116, 120, 129, 132, 150, 151, 165, 173, 183, 203, 205. p<.05 is incorrect. Please, correct (p<0.05).

Response: Correction made

  1. Sections Introduction and Discussion. According to author’s guideline of IJMS, references should be placed after word but before punctuation marks. Please, correct.

Response: Correction made

  1. Table 1. Could you explain, please, why standard deviation values in Amygdala IL-17a are so high (more than average value). Is it correct? Can we trust such data?

Response:  The reviewer is correct that the standard deviation for amygdala IL-17A was quite high. This is not a mistake, and IL-17A gene expression in the amygdala was low overall in all samples.  Few studies have measured IL-17 in the amygdala.  In one study, IL-17A in the amygdala was low in control rats, but significantly increased in diabetic rats (see doi.org/10.1007/s00394-019-01924-7), and in a second study, exposure to stress did not increase IL-17A in the amygdala (see DOI.org/10.1186/s13041-020-00726-x).  Since we do not see a significant effect of stress on IL-17A (and do not suggest that there is a stressor-effect in the manuscript), we have chosen to keep the IL-17A gene expression in the manuscript, but point out that it is highly variable (line 155).

  1. 5, Fig. 6. Please, correct “CxCL1”.

Response: Correction made

  1. line 187 please, delete superfluous "including".

Response: Correction made

  1. line 187 please, correct INF-γ.

Response: Correction made

  1. How can you explain the absence of protein expression of IFN-γ in colon tissue (line 187), whereas its mRNA level was detectable by RT-PCR?

Response:  We agree that it was surprising that IFN-γ was not higher in the colon.  The protein was detectable using the U-Plex assay in 8 out of 18 of the samples that were assessed, but the mRNA was detectable in all of the colons assessed.  PCR methods are highly sensitive, whereas the limit of detection for IFN-γ in the colon was 2.35 pg/mg tissue.  In response to previous comments, we have clarified that the protein was detectable in some, but not all, of the samples and we also state the ratio of samples in each group that were positive.  This is now found in Table 2.   

  1. Table 2. Please, use either MIP-1α or CCL3, but not mix of them.

Response:  We apologize for these errors.  The assay actually assesses MIP-3α (aka, CCL20).  We have switched all occurrences to CCL20.

  1. Section 4.5. The sequences of all PCR-primers used in the study should be added.

Response: We have now included the primer sequences as Supplemental Table 1 in Appendix A.

  1. If you use cytoplex assay to measure protein levels of cyto/chemokines, why was a part of them but not all panel depicted in Fig. 4 ?

Response:  We apologize that we did not make this clear.  We now state that many of the cytokines were below the limit of detection in the serum (lines 175-177) and in the colon (207-213).

Reviewer 3 Report

This interesting manuscript explores the effects of social disruption (SDR) as a stressor following DSS-induced colitis on anxiety-like behavior. SDR increased anxiety-like behavior compared with DSS colitis alone, as well as increased IL-17A and decreased TNF-alpha expression in the hippocampus. Reduced iNOS expression in the colon was associated with upregulated cytokine expression in the mesenteric lymph nodes, potentially connecting inflammation in the colon with that in the brain.

Major:

  1. Figure 1C—confusing graph. Shouldn’t this be plotting the difference in DAI over time (days) between SDR and control groups (i.e., the legend should be SDR and control)? Stress shouldn’t come into play at all until day 9 (and even then, only for the SDR group). Please clarify.
  1. Light/dark exploration (4.3.1) and open field exploration (4.3.2), as well as text associated with these results (lines 112-123)—what values in the measurements indicate anxiety-like behavior? How do your results compare with what the standard values identify as ‘anxiety-like behavior’?
  1. Serum cytokines—the cytokines graphed in Figure 4 don’t match those listed in 4.6. If 4.6 is correct, where are the results for the remaining cytokines?
  1. Need n for all experiments throughout, as well as replicates (technical and biological) for each experiment.
  1. Line 445 (section 4.6), the method applies to both serum and tissue cytokines, correct? If so, address ‘serum’ in title of subsection. How were tissue cytokines scaled to tissue size?

Minor:

  1. Please define what a social disruptor/social defeat stressor is somewhere in the introduction. Authors define the exact protocol in methods, but how is this procedure indicative of anxiety-like behavior? What happens to the mice after this procedure that causes short- or long-term anxiety?
  1. In the introduction, the authors are clear in describing the goals as measuring anxiety during the recovery from mild to moderate DSS-induced colitis. In the abstract, I didn’t catch that the anxiety measurements were during the recovery (even though the days of treatment are mentioned). I would rephrase abstract a bit to ensure reader understands (and you emphasize) these measurements are not concurrent with active IBD, but rather during recovery from the inflammation, potentially leading to a relapse.
  1. Line 73, remove comma before in-text citation and add semi-colon following citation.
  1. Line 76, remove ‘d’ from ‘exacerbated’.
  1. Line 90, remove word ‘mice’ following ‘while’.
  1. For Figure 1, in A, did the behavioral test occur on a day the mice also experienced SDR? I understood from the text that the behavioral test occurred on the day following the last SDR. Please rephrase or redraw to make this more clear.
  1. Figure 3b. TNF-alpha expression is lower in the stress group compared to control, no?
  1. Tables 1 and 2, suggest removing words ‘amygdala’ and ‘colon’ from each row to make reading cytokine/chemokine easier. What does the +/- represent in fold change? SEM? SD? Please clarify.
  1. Line 187, remove second ‘including’.
  1. Line 375, add ‘to’ before ‘or’.
  1. For PCR, please list genes analyzed (section 4.5).
  1. Line 436, add comma following (MLN).
  1. Experimental Design: how was the recovery period between days 5 and 9 determined? Is this a standard period in the literature? If the DSS colitis was mild to moderate, were the mild cases totally resolved at this point?
  1. Would remove IRB section (lines 470-472) as IACUC does not fall under the same guidelines.
  1. Remove informed consent (there was none…not a human study)—lines 473-480.
  1. Remove Data Availability Statement (lines 481-485).
  1. Either remove acknowledgements or personalize (lines 486-488).
  1. Remove Appendix A and B unless you have something to add…(lines 493-502).
  1. Did level of anxiety correlate with intestinal inflammation levels preceding the behavior tests? For example, did mice with mild inflammation show milder levels of anxiety?
  1. Can authors speculate on why TNF-alpha levels in the hippocampus were lower, while they were higher or similar in all other tissues measured in the stressed animals?
  1. Clinically, does anxiety/depression resolve when intestinal inflammation resolves in IBD patients, or are these comorbid conditions longer lasting typically? Is a ‘flare’ of IBD defined only by intestinal symptoms, or can unresolved anxiety/depression also constitute a ‘flare’ by clinical definitions?

Author Response

Reviewer 3 Comments and Suggestions for Authors

This interesting manuscript explores the effects of social disruption (SDR) as a stressor following DSS-induced colitis on anxiety-like behavior. SDR increased anxiety-like behavior compared with DSS colitis alone, as well as increased IL-17A and decreased TNF-alpha expression in the hippocampus. Reduced iNOS expression in the colon was associated with upregulated cytokine expression in the mesenteric lymph nodes, potentially connecting inflammation in the colon with that in the brain.

Major:

  1. Figure 1C—confusing graph. Shouldn’t this be plotting the difference in DAI over time (days) between SDR and control groups (i.e., the legend should be SDR and control)? Stress shouldn’t come into play at all until day 9 (and even then, only for the SDR group). Please clarify.

Response:  We apologize that this was not clear.  The reviewer is correct that the legend should be Stress and Control (which has been fixed).  We have also included a blue bar on days 0-5 to indicate when DSS occurred and a bar on the afternoons of days 9-14 to indicate when the stress exposure occurred.

  1. Light/dark exploration (4.3.1) and open field exploration (4.3.2), as well as text associated with these results (lines 112-123)—what values in the measurements indicate anxiety-like behavior? How do your results compare with what the standard values identify as ‘anxiety-like behavior’?

Response:  There are not established values of what constitutes anxiety-like behavior on these tests.  However, based on the light:dark preference test (which was the only test that showed anxiety-like behavior in stressor-exposed mice), we observed an 18% increase in the amount of time animals spent in the dark portion of the box.  This is in the same range as our previous studies in which we observed a 25% increase (DOI: 10.1016/j.bbi.2015.06.025) and 10% increase (DOI: 10.1523/JNEUROSCI.0450-11.2011) in the amount of time stressor-exposed mice spend in the dark portion of the light dark box.  It is also in the same range observed by Lu et al (2021) which was ~15% increase (estimated based on DOI: 10.1016/j.neuropharm.2021.108816).  We clarify that the increase in the amount of time spent in the dark portion of the light/dark exploration is consistent with previous studies and indicative of anxiety-like behavior on lines 121-123).

  1. Serum cytokines—the cytokines graphed in Figure 4 don’t match those listed in 4.6. If 4.6 is correct, where are the results for the remaining cytokines?

Response:  We apologize that this was confusing. Several cytokines were detectable in some of the samples, but not all of the colon samples. These were inadvertently reported as not-detectable in the text. We have specified this on lines 207-213, and included this information in Table 2.  In addition, IL-17E/IL-25 and IL-21 were inadvertently left out of 4.6, and we now report which cytokines were below the limit of detection in the serum on lines 175-177.

  1. Need for all experiments throughout, as well as replicates (technical and biological) for each experiment.

Response: We now clarify that this was a single experiment with N=18 (9 mice per group) (lines 423, lines 427-428).  We also point out that there were 3 cages of mice in each condition (lines 439-441).

  1. Line 445 (section 4.6), the method applies to both serum and tissue cytokines, correct? If so, address ‘serum’ in title of subsection. How were tissue cytokines scaled to tissue size? Titled adjusted.

Response: The reviewer is correct that cytokine levels in the colon were scaled based on tissue size and are simply expressed as pg/mg of tissue assayed.  This is now corrected in section 4.6.

Minor:

  1. Please define what a social disruptor/social defeat stressor is somewhere in the introduction. Authors define the exact protocol in methods, but how is this procedure indicative of anxiety-like behavior? What happens to the mice after this procedure that causes short- or long-term anxiety?

Response: We have clarified this now on lines 81-84 to state that mice exposed to this stressor are known to have increases in cytokines, such as IL-1 and IL-6, that in turn lead to the development of anxiety-like behavior. 

  1. In the introduction, the authors are clear in describing the goals as measuring anxiety during the recovery from mild to moderate DSS-induced colitis. In the abstract, I didn’t catch that the anxiety measurements were during the recovery (even though the days of treatment are mentioned). I would rephrase abstract a bit to ensure reader understands (and you emphasize) these measurements are not concurrent with active IBD, but rather during recovery from the inflammation, potentially leading to a relapse.

Response:  We have reworked the Abstract, which we believe is now clear that behavioral measurements occurred in mice recovering from mild to moderate colitis.

  1. Line 73, remove comma before in-text citation and add semi-colon following citation.

Response: Corrected throughout

  1. Line 76, remove ‘d’ from ‘exacerbated’.

Repsonse: Corrected

  1. Line 90, remove word ‘mice’ following ‘while’.

Response:  This has been corrected

  1. For Figure 1, in A, did the behavioral test occur on a day the mice also experienced SDR? I understood from the text that the behavioral test occurred on the day following the last SDR. Please rephrase or redraw to make this more clear.

Response:  The reviewer is correct that the behavioral test occurred the morning following the last cycle of SDR. We have clarified this on lines 434-435 and have redrawn Figure 1A to indicate that SDR was in the afternoon and behavioral testing did not occur on the same day as exposure to SDR.

  1. Figure 3b. TNF-alpha expression is lower in the stress group compared to control, no?

Response: That is correct that TNF mRNA was lower in the hippocampus of mice exposed to stress.  It is not yet clear why this occurred, since TNF is typically increased in mice exposed to social stressors and acute DSS has been shown to increase hippocampal TNF.  However, see our response to point 20 below for speculation on why TNF may be lower.  This observation is interesting and warrants further investigation, but that is outside the scope of the current study. 

  1. Tables 1 and 2, suggest removing words ‘amygdala’ and ‘colon’ from each row to make reading cytokine/chemokine easier. What does the +/- represent in fold change? SEM? SD? Please clarify.

Response: We have modified the table as suggested and included a legend indicating that the +/- is the standard error of the mean.   

  1. Line 187, remove second ‘including’.

Response: This has been corrected. 

  1. Line 375, add ‘to’ before ‘or’.

Response:  We have included the word to before the word or as suggested.

  1. For PCR, please list genes analyzed (section 4.5). add to manuscript or supplemental.

Response:  As suggested, we have included a list of genes analyzed (as well as their primer sequences) to the Supplemental Table.

  1. Line 436, add comma following (MLN).

Response:  This has been added.

  1. Experimental Design: how was the recovery period between days 5 and 9 determined? Is this a standard period in the literature? If the DSS colitis was mild to moderate, were the mild cases totally resolved at this point?

            Response:  We conducted pilot experiments based on findings from doi:/full/10.1080/19490976.2017.1372077 that suggested that colitis was increased 3 days after terminating DSS administration.  In our pilot studies, we found that we could consistently obtain mild-moderate colonic inflammation that was reduced down to a mild colitis 3-4 days after terminating DSS.  This allowed us to test whether exposure to stress at a period when colonic inflammation was low would lead to a reactivation of colonic inflammation.  Although the stress did not reactivate colonic inflammation, it did lead to higher anxiety-like behavior and extra-intestinal inflammation that we believe has important implications and forms the basis for additional follow-up studies.

  1. Would remove IRB section (lines 470-472) as IACUC does not fall under the same guidelines.

Response: Thank you for pointing this out, we have now removed this section.

  1. Remove informed consent (there was none…not a human study)—lines 473-480.

Response: This has now been corrected.

  1. Remove Data Availability Statement (lines 481-485).

Response:  In response to the other reviewers, we now state that Data are available upon request.

  1. Either remove acknowledgements or personalize (lines 486-488).

Response: This has now been corrected.

  1. Remove Appendix A and B unless you have something to add…(lines 493-502).

Response: Thank you for pointing this out.  As a note, we have added two supplementary tables that are now in Appendix A.

  1. Did level of anxiety correlate with intestinal inflammation levels preceding the behavior tests? For example, did mice with mild inflammation show milder levels of anxiety?

Response: We tested anxiety-like behavior one day prior to assessing colonic inflammation.  There were no correlations between anxiety-like behavior and hallmarks of colonic inflammation (i.e., histopathology scores and colonic TNF-α).  As shown in the network plot, the primary indicator of anxiety-like behavior (i.e., the amount of time spent in the dark (LD_Dark)) was only correlated with iNOS mRNA in the colon, but this was a negative correlation.  There were no other direct correlations between measures of colonic inflammation and anxiety-like behavior.  In contrast, anxiety-like behavior was more strongly related to extra-intestinal inflammation, including inflammation in the brain and blood.  Thus, it is evident that animals with lower levels of extra-intestinal inflammation showed milder levels of anxiety, but this was not directly correlated to measures of inflammation in the colon.

  1. Can authors speculate on why TNF-alpha levels in the hippocampus were lower, while they were higher or similar in all other tissues measured in the stressed animals?

Response:  This is an interesting question, and we do not yet know why TNF was lower in the hippocampus.  However, we are happy to speculate as asked by the reviewer.  Both chronic DSS and social defeat stressor have been shown to increase TNF-α in the hippocampus.  However, there are multiple studies suggesting that susceptibility and resilience to chronic stress-induced inflammation is influenced by bacteria (see for example www.pnas.org/cgi/di/10.1073/pnas.1600324113), and that administering LPS prior to exposure to chronic social defeat stress prevents stress-induced increases in hippocampal TNFα (see for example doi.org/10.1016/j.bbi.2020.11.002 and doi.org/10.1016/j.neuropharm.2021.108816).  Thus, it is possible that alterations in the microbiome due to administration of DSS reduces hippocampal TNF-α when mice are subsequently exposed to a social defeat stressor.  While interesting, this is only speculation at this point, and thus, we have not included this in the Discussion of the manuscript.

  1. Clinically, does anxiety/depression resolve when intestinal inflammation resolves in IBD patients, or are these comorbid conditions longer lasting typically? Is a ‘flare’ of IBD defined only by intestinal symptoms, or can unresolved anxiety/depression also constitute a ‘flare’ by clinical definitions?

Response: From a clinical perspective, we typically define a flare of IBD by intestinal symptoms, including abdominal pain, diarrhea, weight loss, or changes in laboratory tests (such as increases in inflammatory markers like CRP and sedimentation rate) (lines 283-284). Currently, increases in anxiety or depression are not used to define a flare.  Although this is an interesting concept, we also recognize that anxiety and depression is very common in patients with IBD, even when their disease activity is low.  Thus, anxiety and depression, per se, do not perfectly map onto IBD disease severity and thus, is unlikely to be a useful marker of a flare of disease activity.  In contrast, extra-intestinal cytokines that are increased during stress, such as IL-17A, may be a target for improving mental health in patients with IBD (discussed as a possibility on lines 398-408).

Round 2

Reviewer 1 Report

Thank you for the revisions.

Reviewer 3 Report

Much improved--just a few minor things to fix following. The title for section 4.6 still says Serum cytokines, when both serum and tissue cytokines are measured.

In Figure 3B, asked about TNF because the caption says it is higher in the stress group, but the authors have indicated (and figure shows), TNF is lower in the stress group. Please fix this error.

Authors mention IACUC protocol before references, but in the IRB section. IACUC is not the same as IRB, and need only be included in the methods.